# Cranial Vault Deformation and Its Association with Mandibular Deviation in Patients with Facial Asymmetry: A CT-Based Study

**DOI:** 10.3390/diagnostics15131702

**Published:** 2025-07-03

**Authors:** Mayuko Nishimura, Chie Tachiki, Taiki Morikawa, Dai Ariizumi, Satoru Matsunaga, Keisuke Sugahara, Yasuo Aihara, Akira Watanabe, Akira Katakura, Takakazu Kawamata, Yasushi Nishii

**Affiliations:** 1Department of Orthodontics, Tokyo Dental College, Chiyoda, Tokyo 101-0061, Japan; mayukoa75@gmail.com (M.N.);; 2Department of Anatomy, Tokyo Dental College, Chiyoda, Tokyo 101-0061, Japan; 3Department of Oral Pathobiological Science and Surgery, Tokyo Dental College, Chiyoda, Tokyo 101-0061, Japan; 4Department of Neurosurgery, Tokyo Women’s Medical University, Shinjuku, Tokyo 162-8666, Japan; 5Department of Oral & Maxillofacial Surgery, Tokyo Dental College, Chiyoda, Tokyo 101-0061, Japan

**Keywords:** craniofacial morphology, computed tomography, geometric morphometrics

## Abstract

**Background/Objectives**: Facial asymmetry is known to cause not only deformities in the facial skeleton but also alterations in the cranial vault. However, limited research has explored the association between mandibular asymmetry and cranial vault deformation. This study aimed to evaluate the three-dimensional craniofacial morphology, including the cranial vault, in patients with skeletal mandibular prognathism using computed tomography (CT) imaging. **Methods**: Patients were classified into two groups: those with facial asymmetry (ANB ≤ 0°, Menton deviation ≥ 4 mm) and those without (ANB ≤ 0°, Menton deviation < 3 mm). Reference planes were established in three orthogonal directions, and distances from anatomical landmarks on the maxilla and mandible to each reference plane were measured. Additionally, the cranial vault was segmented into four regions, and the volume of each section was calculated. **Results**: Compared with the symmetric group, the asymmetric group exhibited significant lateral displacement in the maxilla and both anteroposterior and lateral displacements in the mandible. Furthermore, a significant difference in the posterior cranial vault volume between the left and right sides was observed in the asymmetric group. A moderate positive correlation (r = 0.543, *p* = 0.045) was also found between the direction of mandibular deviation and the direction of posterior cranial vault deformation. **Conclusions**: A moderate positive correlation (r = 0.543, *p* = 0.045) was observed between mandibular deviation and posterior cranial vault asymmetry. These findings may suggest that the cranial vault morphology may influence facial asymmetry, and it may serve as one of the considerations for discussing the necessity of early intervention for cranial vault deformity during infancy.

## 1. Introduction

Approximately 70% of patients diagnosed with skeletal jaw deformity are diagnosed with skeletal mandibular protrusion [1]. Tani et al. reported that facial asymmetry was observed in 21–74% of patients diagnosed with mandibular deformity accompanied by the protrusion or retrusion of the maxilla and mandible [2,3,4]. Approximately 50% of patients with skeletal mandibular prognathism have facial asymmetry [5].

In patients with skeletal mandibular prognathism and facial asymmetry, significant bilateral differences have been reported in the mandibular body length, mandibular ramus height, mandibular condylar length, and condylar volume, suggesting that the overall morphology of the mandible influences facial asymmetry [6,7,8]. In addition, the lateral movement of the mandible due to rotational elements such as yaw, pitch, and roll has been reported in some cases [9]. Moreover, it has been suggested that the three-dimensional position of the mandibular fossa is deeply involved in facial asymmetry, indicating the possibility that the mandible may develop asymmetry due to various causes [10]. In addition, cases of facial asymmetry accompanied by maxillary deviation have been reported, and it is essential to evaluate both the maxilla and mandible for diagnosis [11]. Moreover, reports suggest that deformities in patients with facial asymmetry may extend to the cranial base and cranial vault [12,13].

The relationship between mandibular deviation and cranial vault deformity remains unclear. While there are reports analyzing these deviations using axial cephalometric radiographs in two dimensions [14], there are almost no studies analyzing them in three dimensions. In this study, we measured CT images of patients diagnosed with skeletal mandibular protrusion, including those with and without facial asymmetry, and compared the degrees of deviation of the maxilla and mandible. Moreover, we investigated the relationship between the direction of mandibular deviation and cranial vault deformation by dividing the cranial vault into four sections and measuring and comparing their volumes.

## 2. Materials and Methods

### 2.1. Research Subjects and Materials

The subjects of this study consisted of 25 patients diagnosed with skeletal mandibular prognathism accompanied by facial asymmetry (asymmetry group) and 25 patients with skeletal mandibular prognathism without facial asymmetry (symmetry group). All patients received surgical orthodontic treatment at the Department of Orthodontics, Tokyo Dental College Chiba Dental Center (formerly Chiba Hospital Orthodontic Department) between July 2010 and March 2020. The inclusion criteria required all 50 subjects to present an ANB angle (ANB: anterior–posterior relationship of the maxilla and mandible) of less than 0° and a Wits appraisal (Wits: relative position of the maxilla and mandible) of less than 0 mm. Additionally, the Menton (the extreme inferior point of the chin) deviation was defined as ≥4 mm in the asymmetry group and <3 mm in the symmetry group. Patients with hereditary diseases, congenital disorders, endocrine or metabolic abnormalities, a history of craniofacial trauma, or severe TMJ disorders were excluded from this study. A breakdown of the study population is presented in Table 1.

The required sample size was determined based on a preliminary study. The effect size was calculated using the mean and standard deviation of the volume difference between the left and right sides of the posterior cranial vault. The differences were 2488.98 ± 1987.10 mm^3^ for the symmetry group and 4752.44 ± 3279.98 mm^3^ for the asymmetry group. Assuming a significance level of 0.05, a power of 0.75, and a two-tailed test, the estimated sample size was determined to be 21. This decision was further supported by a previous study that investigated facial asymmetry in patients with skeletal mandibular prognathism using three-dimensional analysis, which employed a sample size of 21 [15].

Pre-orthodontic treatment X-ray CT images were used to measure the maxillofacial and cranial morphology. X-ray CT imaging was performed using the Somatom Plus 4 Volume Zoom^®^ and a Somatom Definition AS^®^ (Siemens, Erlangen, Germany). The scanning parameters were as follows: X-ray tube voltage, 120 kV; X-ray tube current, 117 or 88 mA; slice thickness, 1.25 or 1.00 mm; and slice interval, 1.0 mm. All image data were extracted in DICOM format. The extents of the maxillary and mandibular deviation and cranial vault volume were measured using SimPlant Pro^®^ 2011 (Materialise Dental, Leuven, Belgium), a maxillofacial treatment simulation software. All three-dimensional measurements and analyses in this study were performed by the same operator. Four weeks later, measurements were repeated at the reference points comprising the reference planes, and then the intraclass correlation coefficient was calculated (ICC; 1, 0, 0.84).

This study was conducted with the approval of the Ethics Committee of the Tokyo Dental College School of Dentistry (Approval Number: 916).

### 2.2. Establishment of Reference Points, Reference Planes, and Measurement Items in CT Images

In the craniofacial region, including the cranial vault, the horizontal reference plane (X) passing through the bilateral porions (PoR, PoL) and left orbitale (OrL), the mid-sagittal reference plane (Y) passing through the nasion (N) and basion (Ba), which is perpendicular to the horizontal reference plane, and the frontal reference plane (Z) passing through the bilateral foramen spinosum (FsR, FsL), which is perpendicular to the horizontal reference plane, were set as the reference planes. The distances from the reference points of the maxilla (GP: opening in the posterior hard palate of the greater palatine canal) and mandible (Go: most posterior, inferior, and lateral point on the external point at the angle of the mandible) bones to the three reference planes were measured. A plane parallel to the horizontal plane passing through the glabella (G) was set to divide the cranial vault into four sections with the mid-sagittal reference plane and the frontal reference plane. A total of six reference points were established to define the reference planes. In addition, the anatomical structures present on the left and right sides were also used as measurement points on the maxilla (GP) and mandible (Go) (Table 2, Figure 1, Figure 2).

### 2.3. Measurement and Evaluation

#### 2.3.1. Comparison of Mandibular Deviation Between Asymmetric and Symmetric Groups

The mandibular deviation was measured in the asymmetric and symmetric groups. In the maxilla, the distance (mm) from the left and right GP to each reference plane was measured as |RGP − LGP|. In the mandible, the distance (mm) from the left and right Go to each reference plane was measured as |RGo − LGo|. Since the direction of deviation at the measurement points varied among cases, the differences in the left–right distances were calculated as absolute values in the asymmetric and symmetric groups, and the degrees of maxilla and mandibular deviation were compared using |X − GP|, |Y − GP|, |Z − GP|, |X − Go|, |Y − Go|, and |Z − Go|.

#### 2.3.2. Comparison of Cranial Vault Volume Between Asymmetric and Symmetric Groups

The cranial vault was divided into four sections by each reference plane, labeled V1, V2, V3, and V4, and their volumes (mm^3^) were measured. Similar to distance measurement, the direction of deformation of the cranial vault varies depending on the case. Therefore, the left–right differences in the volumes of the frontal and occipital regions in the asymmetrical and symmetrical groups were calculated as |V1 − V2| and |V3 − V4|, respectively, and the degrees of deformation were compared.

#### 2.3.3. Measurement of the Displacement Direction and Amount of Menton in Facial Asymmetry Groups

The asymmetric group was divided into two groups: one group in which the direction of Menton deviation was the same as the direction of occipital cranial deformation, and another group in which the directions were different. The correlation between the amount of Menton deviation and the difference in the cranial volume was examined in the frontal and occipital regions of the cranial vault.

### 2.4. Statistical Analysis

The *t*-test was used to compare the asymmetric and symmetric groups in terms of the left–right differences in the anteroposterior, vertical, and lateral positions of the maxilla and mandible, as well as in terms of the left–right differences in the cranial vault volume. The normality of the data was assessed using the Shapiro-Wilk test prior to conducting the *t*-test, and the assumption of normality was found to be satisfied. Pearson’s correlation coefficient analysis was used to determine the correlation between the left–right differences in the cranial vault and the degree of mandibular deviation. Statistical analysis was performed using SPSS version 24.0 (IBM Corporation, Armonk, NY, USA).

## 3. Results

### 3.1. Comparison of Distance Measurements of Anatomical Structures in Maxilla and Mandible in Symmetric and Asymmetric Groups

Compared with the facial symmetry group, the facial asymmetry group demonstrated a significantly larger difference in the distance from the left and right GP to the Y-plane in the maxilla. In the mandible, the difference in the distance from the left and right Go to the Y-plane and Z-plane was significantly larger (Table 3, Figure 3).

### 3.2. Comparison of Left–Right Differences in Cranial Vault Volume Measurements Between Symmetric and Asymmetric Facial Groups

Compared with the symmetric group, the asymmetric group demonstrated significantly larger left–right differences in the volume of the posterior cranial vault. No significant differences were observed in the frontal region, but the asymmetric group tended to have larger differences (Figure 4).

### 3.3. Relationship Between Posterior Cranial and Mandibular Deviation

Among the 25 patients with facial asymmetry, 15 exhibited a deviation of the posterior cranial vault on the same side as the direction of the deviation of Menton. In the group in which the direction of Menton deviation and the direction of deformation of the cranial vault were on the same side, a moderate positive correlation (r = 0.543, *p* = 0.045) was observed between the degree of Menton deviation and the difference in the volume of the posterior cranial vault. However, no correlation was found in the group in which they were on different sides. Moreover, no correlation was observed between the degree of Menton deviation and the difference in the volume of the anterior cranial vault (Figure 5 and Figure 6).

## 4. Discussion

### 4.1. Deviation of the Maxilla and Mandible in Patients with Facial Asymmetry

The results of this study demonstrated a significant difference in the distance from the mandibular angle to the frontal reference plane and the sagittal reference plane in the asymmetric group, indicating the anterior–posterior and lateral deviation of the mandibular angle. The area surrounding the mandibular angle is attached to the masseter muscle and the medial pterygoid muscle. Zhao et al. reported that the volume of the masseter muscle in patients with skeletal mandibular protrusion accompanied by facial asymmetry was larger on the non-deviated side compared with that on the deviated side of the mandible [16]. Moreover, Kwon et al. reported that the volume of the medial pterygoid muscle is smaller on the mandibular deviation side [17]. In addition, the mandibular fossa on the mandibular deviation side in patients with facial asymmetry is displaced posteriorly and laterally compared with that on the opposite side [10]. Since growth occurs due to endochondral ossification in the mandibular condyle, the deviation of the mandibular fossa may lead to the deviation of the mandibular condyle position, thereby influencing mandibular asymmetry [18]. Since facial asymmetry is not limited to the mandible but also involves muscles and the mandibular fossa, it is speculated that these structures interact, resulting in the complex deviation of the mandible.

There are reports that the deviation of the maxilla is often simple lateral displacement [19]. In this study, we found a significant difference in the distance from the left and right pterygomaxillary foramen to the mid-sagittal reference plane in the maxilla, supporting similar results. The maxilla is adjacent to the anterior cranial base and exhibits an intermediate growth pattern between the neural and general types, leading to an earlier growth peak compared with that of the mandible [20]. Moreover, the maxilla grows as part of the nasomaxillary complex through sutures with adjacent bones, suggesting that asymmetric growth is less likely to occur compared with the mandible [21]. These factors contribute to the maxilla exhibiting a simpler growth pattern compared with that of the mandible.

### 4.2. Differences in the Volume of the Posterior Cranial Vault

The cranial base, which constitutes the cranial vault, is divided into three parts: the anterior cranial base, the middle cranial base, and the posterior cranial base. In this study, the mid-sagittal plane dividing the cranial vault into anterior and posterior parts was defined based on the sphenoid foramen located on the greater wing of the sphenoid bone. Since the sphenoid foramen is located anterior and medial to the mandibular fossa, the majority of the temporal bone, including the mandibular fossa, is located within the middle cranial base. In this study, the middle cranial base was included in the posterior cranial region of the cranial vault [22]. Hayashi et al. reported that patients with facial asymmetry exhibit a greater volume difference between the left and right sides of the middle cranial base compared with the anterior and posterior cranial bases [13]. This result aligns with our findings, which demonstrated a significant difference in the volumes of the left and right sides of the posterior cranial vault between the asymmetric and symmetric groups. Deformation in the temporal bone located in the middle cranial base is thought to affect the position of the mandibular fossa and cause the lateral deviation of the mandible. Moreover, while the growth of the cranial vault is classified as a neural growth pattern with a growth peak around age 6, the mandibular fossa continues to grow until around age 12. Therefore, it is reasonable to assume that deformation in the cranial vault may contribute to mandibular fossa deviation [20,23].

### 4.3. Correlation Between Mandibular Deviation and Cranial Vault Volume

In this study, approximately 60% of cases in the asymmetric group had mandibular deviation and occipital deformation in the same direction, and a moderate positive correlation was observed between the degree of mandibular deviation and the degree of occipital deformation. This finding is consistent with the report by Kawamura et al., who reported that as mandibular asymmetry increases, the mandibular fossa on the side of the mandibular deviation shifts more posteriorly and inferiorly [24]. The posterior cranial region in this study includes the middle cranial base and the posterior cranial base. The ossification of the sphenobasilar synchondrosis, which is in the middle cranial base and posterior cranial base, is completed around the age of 18 to 20, similar to the completion of the growth of the mandibular fossa [25]. According to Enlow’s theory of growth equilibrium, the growth of sutures and synchondroses in the craniofacial region is said to be correlated [20]. Deformation of the temporal bone can affect other bones that constitute the cranial vault through sutures and cartilaginous joints. Moreover, the deviation of the mandibular fossa, which is composed of the temporal bone, can cause the deviation of the mandible. Therefore, it is speculated that these deformations and deviations tend to occur on the same side due to mutual influence.

### 4.4. Research Methods

The American Association of Oral and Maxillofacial Surgeons defines facial asymmetry as a lateral deviation of Menton by 3.0 mm or more [26]. Moreover, other studies recognize criteria where the lateral deviation of Menton is approximately 4 mm or more as a standard for distinguishing the presence or absence of facial asymmetry [27,28]. Based on this, in this study, cases with a lateral deviation of Menton of 4 mm or more were classified as the asymmetric group, and those with less than 3 mm were classified as the symmetric group.

Cephalometric analysis is a common method for determining treatment plans for orthodontic treatment and orthognathic surgery. Frontal cephalometric X-ray photographs are commonly used to evaluate facial asymmetry. However, due to the overlapping of maxillofacial structures in the anteroposterior plane, it is challenging to accurately assess the complex three-dimensional morphology of the maxillofacial skeleton using two-dimensional images [29]. As a result, three-dimensional imaging has increasingly been employed in recent years, particularly in surgical orthodontic treatment, with approximately 40% of patients undergoing preoperative diagnosis using three-dimensional imaging prior to orthognathic surgery [1].

CT images allow for more accurate assessments of the teeth and jawbones compared with conventional X-ray photographs. However, no reference planes have been defined. In this study, we established reference points for defining the reference planes: the horizontal reference plane uses the porion and orbitale, the mid-sagittal reference plane uses the nasion and basion, and the frontal reference plane uses the foramen spinosum. These landmarks were adopted in this study due to their high reproducibility [30,31]. Moreover, the foramen spinosum was selected as the reference point for the mid-sagittal reference plane because it is considered to have a stable shape unaffected by deformities of the surrounding bones [32].

### 4.5. Positional Plagiocephaly

Deformities of the skull in infancy have recently attracted significant attention. In the United States, to reduce sudden infant death syndrome (SIDS), the American Academy of Pediatrics recommended supine sleeping in 1992 and launched the “Back to Sleep” campaign in 1994. While this reduced the incidence of SIDS, the prevalence of positional plagiocephaly increased dramatically from 0.3% to 48% [33]. Cranial vault deviations are classified into dolichocephaly, brachycephaly, and plagiocephaly, with plagiocephaly being most closely associated with facial asymmetry [12]. Position-related plagiocephaly is a condition where part of the cranial vault becomes flattened due to postnatal factors, and it is the most common type of cranial vault deformation. In Japan, the incidence of plagiocephaly at 6 months is 45.1%, that of brachycephaly is 22.0%, and that of dolichocephaly is low. Reports indicate that plagiocephaly is more commonly observed in preterm infants in Japan [34,35]. Plagiocephaly not only causes craniofacial deformities but also increases the risk of strabismus, otitis media, and delayed psychomotor development; as a result, early intervention for cranial vault deformities is increasing [36,37,38,39]. Representative evaluation items for cranial vault deformities include the cranial asymmetry (CA) and cranial vault asymmetry index (CVAI). These evaluation criteria are used to classify the severity of cranial vault deformities. In positional plagiocephaly, moderate to severe cases often require treatment [40,41]. Facial asymmetry is rarely a problem in infancy but becomes more pronounced during puberty and can lead to psychological issues. If cranial vault deformation in infancy is considered a contributing factor to facial asymmetry, it is anticipated that the number of patients seeking treatment for cranial vault deformation in infancy will increase in the future. In this study, a correlation was observed between the degree of mandibular deviation and the cranial vault deformation, and the deformation directions tended to be ipsilateral. Since cranial deformation is recognized shortly after birth, it is possible that cranial deformation may lead to the deformation of the midface and lower face; this should be addressed in future research.

### 4.6. Limitations

This study has several limitations that should be acknowledged. First, due to its retrospective and cross-sectional design, causal relationships cannot be established. Second, although the sample size was balanced between the groups, the overall number of participants was relatively small. Third, as the CT data were obtained from a specific population, the generalizability of the findings may be limited. Future studies with larger, more diverse cohorts and longitudinal designs are recommended to validate and expand upon these findings.

### 4.7. Clinical Implication

Based on the results of this study, early intervention for cranial vault deformation in infancy may serve as a potential strategy to prevent facial asymmetry during subsequent craniofacial development. Further studies are warranted to investigate whether preventing cranial deformation in early life can contribute to reducing the risk of facial asymmetry in the long term.

## 5. Conclusions

In cases with facial asymmetry, three-dimensional analysis of the maxilla, mandible, and cranial vault revealed the lateral deviation of the maxilla and lateral and the anteroposterior deviation of the mandible. Moreover, there was a significant difference in the volume of the posterior cranial vault, and there was a tendency for the deviation of the mandible and the deformation of the cranial vault to be ipsilateral, with a correlation observed in the degrees of these findings.

## Figures and Tables

**Figure 1 diagnostics-15-01702-f001:**
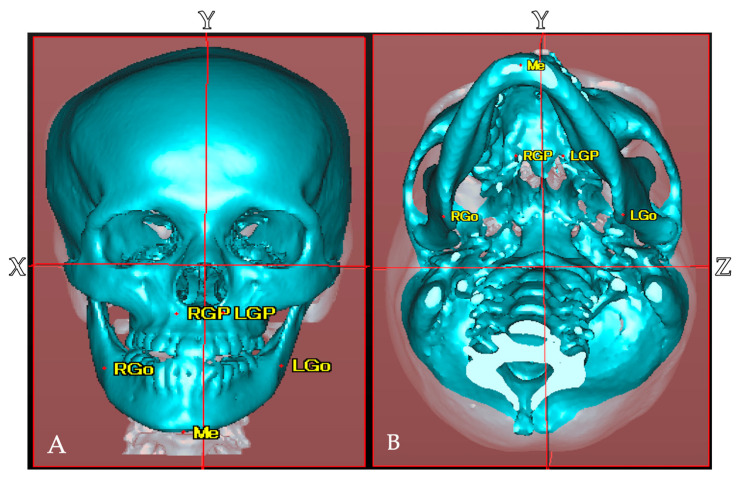
Reference planes and anatomical landmarks. *(***A**) Frontal 3D reconstruction from medical CT data. Horizontal reference plane (X-plane); mid-sagittal reference plane (Y-plane). Anatomical landmarks include the greater palatine foramen (RGP, LGP), gonion (RGo, LGo), and Menton (Me). (**B**) Inferior view demonstrating the frontal reference plane (Z-plane).

**Figure 2 diagnostics-15-01702-f002:**
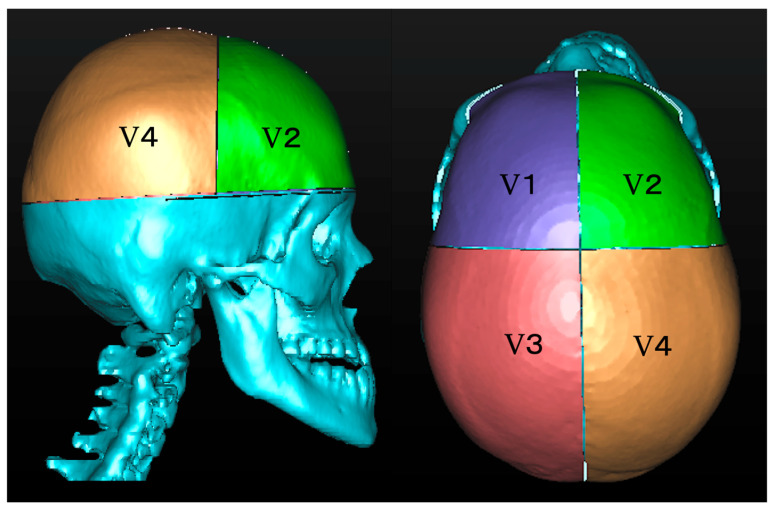
Segmentation of the cranial vault. The cranial vault was segmented using a plane parallel to the horizontal reference plane that passes through the glabella. The segmented cranial vault was further divided into anterior (V1, V2) and posterior (V3, V4) regions along the mid-sagittal plane.

**Figure 3 diagnostics-15-01702-f003:**
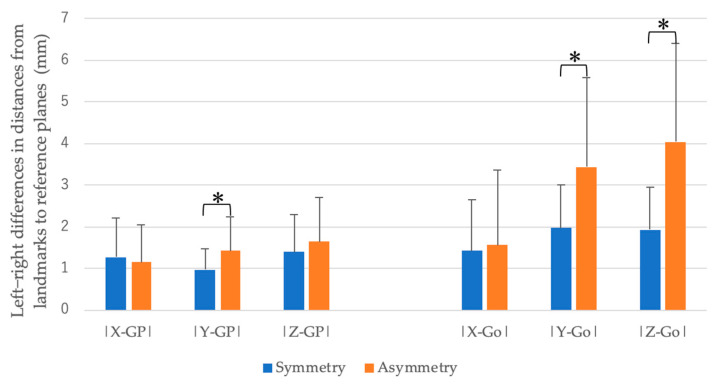
The three-dimensional asymmetry of the maxilla and mandible between groups (** p* < 0.05). The vertical axis indicates the absolute difference in the distances between the bilateral reference points (GP and Go) to each reference plane in the symmetric and asymmetric groups (mm).

**Figure 4 diagnostics-15-01702-f004:**
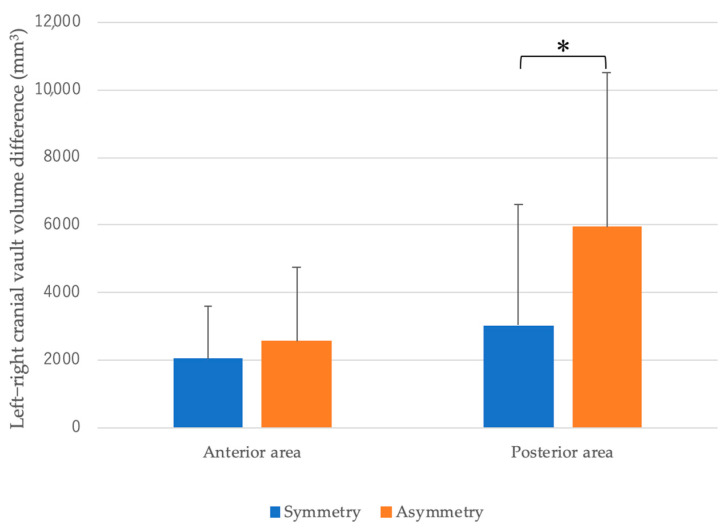
Comparison of anterior and posterior cranial vault asymmetry between groups (** p* < 0.05). The vertical axis shows the absolute differences in the left–right volume for the anterior (|V1 − V2|) and posterior (|V3 − V4|) cranial vault regions in the symmetric and asymmetric groups (mm^3^).

**Figure 5 diagnostics-15-01702-f005:**
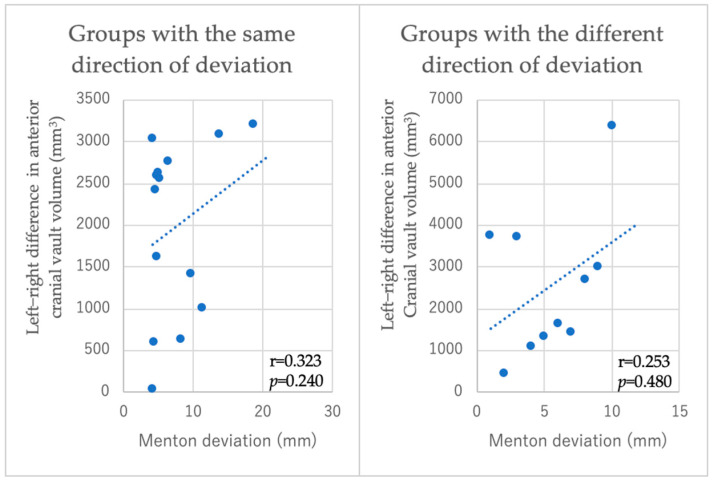
Correlation between anterior cranial vault deformation and Menton deviation. Vertical axis: difference in the anterior cranial vault volume between left and right sides (mm^3^). Horizontal axis: Menton deviation (mm). No significant correlation was observed (|r| = 0.253–0.323, *p* > 0.05).

**Figure 6 diagnostics-15-01702-f006:**
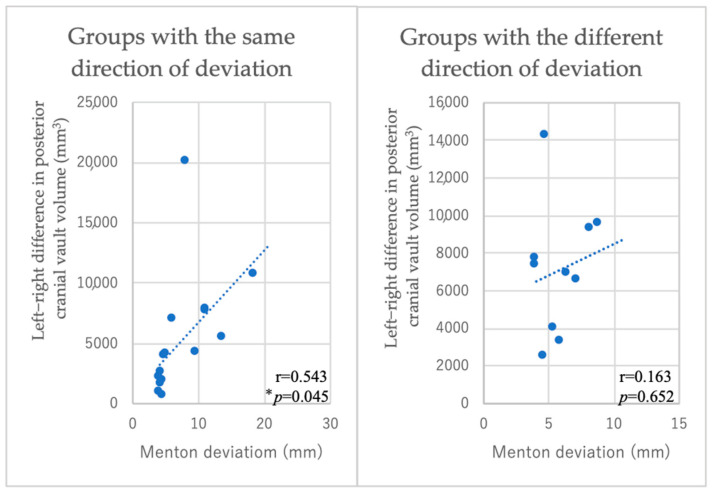
Correlation between posterior cranial vault deformation and Menton deviation (** p* < 0.05). Vertical axis: difference in posterior cranial vault volume between left and right sides (mm^3^). Horizontal axis: Menton deviation (mm). A moderate positive correlation was observed (|r| = 0.543, *p* = 0.045).

**Table 1 diagnostics-15-01702-t001:** Characteristics of patients with skeletal mandible prognathism with and without asymmetry included in this study (*n* = 50).

	Symmetry	Asymmetry
	*n* = 25	*n* = 25
Sex		
Male	9	11
Female	16	14
Age (y)		
Mean	25.7 ± 7.4	25.2 ± 8.6
Range	16.1–45.1	17.2–45.1
Measurement values		
Mean of ANB (°)	−2.8 ± 2.3	−3.7 ± 2.8
Mean of Difference in position of Menton (mm)	1.5 ± 0.68	7.2 ± 3.8
Range of Menton (mm)	0.36–2.89	4.1–18.56
Mean of FMA(°)	28.2 ± 3.9	31.3 ± 3.6
Mean of Wits appraisal (mm)	−8.7 ± 2.7	−11.1 ± 3.5

**Table 2 diagnostics-15-01702-t002:** Definitions of the reference and measurement points.

	Abbreviation	Explanation
Reference point		
Glabella	G	Most prominent point between the eyebrows
Nasion	Na	Most anterior point of the frontal nasal suture
Orbirale	Or	Lowest point of the orbital bone margin
Porion	Po	Sublingual neural tube opening
Foramen spinosum	Fs	Foramen spinosum opening
Basion	Ba	Lowest point on the anterior margin of the foramen magnum
Menton	Me	Lowest point of the mandibular symphysis
Measurement point		
Maxillary		
Greater palatine foramen	GP	Opening in the posterior hard palate of the greater palatine canal
Mandible		
Gonion	Go	Most posterior, inferior, and lateral point on the external point at the angle of the mandible

**Table 3 diagnostics-15-01702-t003:** The measurement of the 3D distances for the corresponding areas on the left and right sides of the maxilla and mandible (mm).

	Maximum (mm)	Minimum (mm)	Mean (mm)
Symmetry			
maxillary			
|X − GP|	3.35	0	1.20
|Y − GP|	2.13	0.22	0.96
|Z − GP|	3.25	0.41	1.01
mandible			
|X − Go|	4.25	0.02	1.62
|Y − Go|	1.26	0.42	1.99
|Z − Go|	3.84	0.61	2.04
Asymmetry			
maxillary			
|X − GP|	3.17	0.01	1.04
|Y − GP|	2.97	0.37	1.38
|Z − GP|	4.08	0.01	1.51
mandible			
|X − Go|	8.15	0.06	1.47
|Y − Go|	9.09	0.41	3.53
|Z − Go|	9.84	1.43	4.02

## Data Availability

The data presented in this study are available upon request from the corresponding author.

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
