# Peer review of "Cranial Vault Deformation and Its Association with Mandibular Deviation in Patients with Facial Asymmetry: A CT-Based Study"

_diagnostics, 2025, doi:10.3390/diagnostics15131702_

Round 1

Reviewer 1 Report

Comments and Suggestions for Authors

Peer Review

Article: Cranial Vault Deformation and Its Association with Mandibular Deviation in Facial Asymmetry Patients: A CT-Based Study

1. Overview

The topic addressed is extremely interesting and little developed in the specialized literature: the link between existing deformities at the level of the skull and mandibular deviation. Using three-dimensional CT allows precise measurements, and the correlation between skull deformation and mandibular deviation in patients with facial asymmetry. The use of three-dimensional CT to establish correlations between the direction of skull deformation and chin deviation, represents an original element of the study.

Three-dimensional measurements and the inclusion of the skull volume as an analysis variable are solid elements of the study.

The study is well structured with an introduction that supports the study.

The methodology is clearly described, and the discussion is relevant.

The results are logically presented and support the hypotheses formulated.

I suggest that in the abstract, the phrase “These findings suggest a close relationship…” could be reformulated (“These findings may suggest…”), since the correlation is moderate.

2. Methodology

The methodology is reliable, and the selection of inclusion/exclusion criteria is justified. To meet the objectives of the study, the cranial skull was divided into four regions and well-defined reference planes were used. The retrospective study is well-controlled, with equalized groups.

I would emphasize as a strength of the study the detailed description of the reference planes and anatomical points used in the study.

3. Statistical analysis

The tests used (t-test and Pearson coefficient) are appropriate. The p-value and the correlations obtained are clearly mentioned. However, I would recommend reporting r-values ​​with Cohen’s interpretation (0.3 – weak, 0.5 – moderate, etc.).

4. Discussion and conclusions

The discussion is extensive and uses relevant references, the results are discussed correctly in relation to the literature.

The conclusions are adequate, but could be more reserved in expression. Thus I would suggest that it should be mentioned that a causal relationship cannot be established, the study being cross-sectional, and I would also introduce a separate section on the limitations of the study.

Recommendation is for acceptance after minimal revisions.

Reviewer 2 Report

Comments and Suggestions for Authors

Overall Assessment

I found this study quite interesting - the authors tackle the relationship between cranial vault deformation and mandibular deviation in facial asymmetry patients using 3D CT, which hasn't received much attention in our field. While the basic methodology is solid and the topic has clear clinical relevance, there are several areas that need work before publication.

The paper reads well overall, but I noticed some issues with precision in the writing and a few methodological concerns that should be addressed.

Detailed Comments

Abstract

Your abstract hits the main points, but the conclusion feels too vague. Instead of saying there's "a close relationship" and mentioning "comprehensive assessment," tell us what you actually found. Give us the numbers - what was the correlation coefficient? What's the practical takeaway for clinicians?

Introduction

I like how you've set up the clinical context and made the case for why this work matters. The prevalence data on mandibular prognathism is well-presented. However, your final paragraph could be sharper - I had to read it twice to understand exactly what you were planning to investigate. A clearer research question would help.

Methods

The patient selection criteria make sense, and I appreciate the detail on imaging protocols. Using ICC for reliability was the right call.

My main concern here is the statistics. Did you check for normality before running those t-tests? I didn't see this mentioned, but it's pretty important given your sample size. Also, while you mention sample size justification, I'd like to see more detail on how you arrived at your numbers.

The CT parameter description gets a bit repetitive - you could tighten this up without losing important information.

Results

The data presentation is generally clear, and your tables are well-organized. I can see the main finding about posterior cranial vault asymmetry correlating with mandibular deviation.

But please be more precise with your language. When you say something "tended to show" a relationship, give us the actual p-value and effect size. "Moderate correlation" needs to come with the correlation coefficient. Your figures (3-6) need better labeling - some of the axes are hard to read, and significance levels aren't always clear.

Discussion

This section works well - you've done a good job connecting your findings to existing literature, and the anatomical reasoning makes sense. The positional plagiocephaly discussion adds useful clinical context.

However, you're not addressing the study's limitations adequately. Your sample size is relatively small, you're working with data from just one institution, and you haven't stratified by growth stage - all of these could affect how broadly we can apply your findings. You also need to be more careful about distinguishing between what your data actually shows versus what you're inferring about underlying mechanisms.

Conclusion

This section needs the most work. Right now it's too generic. How might your findings actually change clinical practice? Should orthodontists be looking at cranial vault morphology more carefully? Does this affect surgical planning? Give us something concrete to take away.

Technical Issues

I spotted a few things that need fixing:

  • "Charasteristics" should be "Characteristics" in Table 1
  • There's a "This this result" duplication somewhere
  • Define your abbreviations (GP, Go, ANB, Wits) on first use
  • Figure captions should stand alone - I shouldn't need to flip back to the text to understand them

Recommendation

Minor Revision Required

This is solid work addressing an important clinical question, and your methodology is generally sound. The findings are novel and potentially useful for practitioners. With the revisions I've outlined - particularly strengthening the conclusion, addressing limitations more thoroughly, and improving precision throughout - this should be suitable for publication.

The core contribution is valuable; it just needs some polish to meet publication standards.
